# Neonatal Drug Formularies—A Global Scope

**DOI:** 10.3390/children10050848

**Published:** 2023-05-08

**Authors:** Dotan Shaniv, Srinivas Bolisetty, Thomas E. Young, Barry Mangum, Sean Ainsworth, Linda Elbers, Petra Schultz, Melanie Cucchi, Saskia N. de Wildt, Tjitske M. van der Zanden, Neil Caldwell, Anne Smits, Karel Allegaert

**Affiliations:** 1Pharmacy Services, Kaplan Medical Center (Clalit Health Services), Pasternak St., P.O. Box 1, Rehovot 76100, Israel; dotan.shaniv@mail.huji.ac.il; 2Neonatal Intensive Care Unit, Kaplan Medical Center (Clalit Health Services), Pasternak St., P.O. Box 1, Rehovot 76100, Israel; 3Division of Newborn Services, Royal Hospital for Women, Sydney 2031, Australia; srinivas.bolisetty@health.nsw.gov.au; 4Co-Founder of Neofax and Acorn Publishing, Raleigh, NC 27610, USA; teyoungmd@gmail.com (T.E.Y.); barrymangum2@gmail.com (B.M.); 5WakeMed Health and Hospitals, Raleigh, NC 27610, USA; 6School of Medicine, University of North Carolina, Chapel Hill, NC 27599, USA; 7Co-Founder and CEO of Paidion Research Inc., Durham, NC 27707, USA; 8School of Medicine, Duke University, Durham, NC 27710, USA; 9Directorate of Women and Children’s Health, Victoria Hospital, Kirkcaldy, Scotland KY2 5AH, UK; sean.ainsworth@nhs.scot; 10Micromedex/NeoFax Editorial Services, Merative, 100 Phoenix Drive, Ann Arbor, MI 48108, USA; lelbers@merative.com (L.E.); pschultz@merative.com (P.S.); 11Wolters Kluwer-Health, Clinical Effectiveness, Waltham, MA 02453, USA; melanie.cucchi@wolterskluwer.com; 12Department of Pharmacology and Toxicology, Radboud University Medical Center, 6525 GA Nijmegen, The Netherlands; saskia.dewildt@radboudumc.nl (S.N.d.W.); t.vanderzanden@erasmusmc.nl (T.M.v.d.Z.); 13Intensive Care and Department of Pediatric Surgery, Erasmus MC-Sophia Children’s Hospital, 3015 CN Rotterdam, The Netherlands; 14Dutch Knowledge Center Pharmacotherapy for Children, Postal Box 25270, 3001 HG Rotterdam, The Netherlands; 15Department of Pediatrics, Erasmus MC Sophia Children’s Hospital, 3015 CN Rotterdam, The Netherlands; 16Pharmacy Department, Wirral University Teaching Hospital/Liverpool John Moores University, Arrowe Park Road, Upton, Merseyside CH49 5PE, UK; neil.caldwell@nhs.net; 17Department of Development and Regeneration, KU Leuven, 3000 Leuven, Belgium; anne.smits@uzleuven.be; 18Neonatal Intensive Care Unit, University Hospitals Leuven, 3000 Leuven, Belgium; 19Department of Pharmaceutical and Pharmacological Sciences, KU Leuven, 3000 Leuven, Belgium; 20Department of Hospital Pharmacy, Erasmus MC, 3015 GD Rotterdam, The Netherlands

**Keywords:** drug information, drug database, drug formulary, neonatal, pediatric

## Abstract

Neonatal drug information (DI) is essential for safe and effective pharmacotherapy in (pre)term neonates. Such information is usually absent from drug labels, making formularies a crucial part of the neonatal clinician’s toolbox. Several formularies exist worldwide, but they have never been fully mapped or compared for content, structure and workflow. The objective of this review was to identify neonatal formularies, explore (dis)similarities, and raise awareness of their existence. Neonatal formularies were identified through self-acquaintance, experts and structured search. A questionnaire was sent to all identified formularies to provide details on formulary function. An original extraction tool was employed to collect DI from the formularies on the 10 most commonly used drugs in pre(term) neonates. Eight different neonatal formularies were identified worldwide (Europe, USA, Australia-New Zealand, Middle East). Six responded to the questionnaire and were compared for structure and content. Each formulary has its own workflow, monograph template and style, and update routine. Focus on certain aspects of DI also varies, as well as the type of initiative and funding. Clinicians should be aware of the various formularies available and their differences in characteristics and content to use them properly for the benefit of their patients.

## 1. Introduction

### 1.1. Drug Information in Neonates

Drug Information (DI) is the collection of chemical and pharmacological characteristics and clinical data pertinent to individual drugs, which clinicians need to know and have access to when considering therapeutic options for a patient. DI is fundamental to making an educated decision about whether that drug is appropriate for the patient and their condition, and which dose to administer. The most clinically useful DI are aspects such as posology (also known as ‘dosing parameters’), warnings and precautions, information regarding formulation and administration, drug interactions and adverse effects (incidence, severity), and pharmacokinetic (PK) and pharmacodynamic (PD) data. Most of these data are provided by the drug’s manufacturer, who obtains it through the pre-clinical and clinical trials required by health authorities to confirm the drug’s safety and efficacy prior to issuing marketing authorization. To make DI accessible to healthcare professionals, carers and patients, the manufacturer summarizes it into several concise forms, including a Patient Information Leaflet (PIL), Summary of Product Characteristics (SmPC) and package labeling. Safety and efficacy data obtained during the post-marketing stage through pharmacovigilance reports, epidemiologic studies and adverse events monitoring systems (e.g., FDA Adverse Event Reporting System (FAERS)) are sometimes also implemented in the drug label as periodic or incidental safety updates [1].

However, the neonatal population, which includes preterm and term infants in the first four weeks of life, may require pharmacotherapy similar to any other population, with two exceptions: drug demand is usually inversely related to the age of the infant, meaning the younger the patient, the more drugs they may require [2]; and DI in this population is usually scarce. Therefore, most drugs routinely prescribed to neonates are not authorized for use in this population, leading to off-label and/or unlicensed use of drugs in neonates. An off-label use essentially means “prescribing of an authorized product for use in a way that is not described in the SmPC” [3]. Unlicensed use also includes using an approved drug differently than its conditions included in the approval, e.g., administering solutions for injections by the oral route, or preparing an extemporaneous oral solution from solid oral dosage forms for infants or children with swallowing difficulties. Several studies have been conducted in an attempt to quantify the off-label prescription rate in neonates. These include a 5-year observational study from Spain by Lizano- Díez et al. [4], which demonstrated that 34.10% of drugs prescribed to neonates over a period of 2 years were prescribed off-label. A prospective cohort study by Costa et al. [5] examined the prevalence of neonates administered off-label and unlicensed drugs in a neonatal intensive care unit (NICU) in Brazil and found that 96.4% of neonates and 100% of extremely preterm neonates or extremely low birth weight neonates were exposed to off-label drugs, while 66.8%, 76.7% and 75.7% were exposed to unlicensed drugs, respectively. This high prevalence has very recently been reconfirmed in a Danish nationwide survey [6]. Moreover, a review by van der Zanden et al. found that 70% and 56% of the drugs included in the Dutch Pediatric Formulary (DPF) are not authorized (i.e., used ‘off-label’) in preterm and term neonates, respectively [7]. This makes neonatal formularies, when based on sound methodology, a valuable and often crucial resource of credible neonatal DI. 

### 1.2. How to Integrate DI into Neonatal Formularies

As neonatal clinical trials intended for drug labeling are still rare, the majority of neonatal DI is based on academic research. Sporadic case reports/case series, observational studies, extrapolation from older children or even adults, and pharmacokinetic data, with or without simulation of doses, further deliver knowledge on neonatal DI. In addition, expert opinions of neonatologists based on pharmacological reasoning and principles of neonatal physiology [8,9] further increase current knowledge on developmental and neonatal DI.

To examine the current ‘bedside’ use of neonatal drug formularies, a questionnaire on aspects related to neonatal pharmacotherapy was sent out to participants of the latest (2022) Neonatal Online Training and Education (NOTE) course on neonatal clinical pharmacology. Fifty-three responses were received from sixty NOTE participants [10]. While limited in number, attendees were working in different regions of the world. This questionnaire included a question on formularies commonly used to assist in the drug dosing decision for neonates. The British National Formulary for Children (BNF for Children) (22 respondents), NeoFax^®^ or Micromedex^®^ (21 respondents), Neonatal Formulary (3 respondents) or Lexicomp through UpToDate (5 respondents) were most commonly mentioned, while 25 colleagues mentioned using local, regional or national formularies. Respondents commonly used several sources of information.

This small sample illustrates the wide and variable use of neonatal formularies. This review aims to map the currently available neonatal formularies in various regions of the world, to describe and compare their practices to process, integrate and implement the ever-growing body of neonatal pharmacotherapy literature into their formularies.

## 2. Materials and Methods

### 2.1. Search Strategy to Identify Neonatal Formularies

To map as many existing neonatal formularies as possible, a pragmatic approach was coordinated by the core group (DS, AS and KA): (a) a structured internet search strategy was employed by using specific keywords (“neonatal”, “formulary” and “database”); (b) experts and colleagues within different professional societies associated with neonatal pharmacotherapy (e.g., European Society for Developmental Perinatal and Paediatric Pharmacology (ESDPPP), International Neonatal Consortium (INC), European Society for Paediatric Research (ESPR), Neonatal Online Training and Education (NOTE), International Postgraduate Organization for Knowledge transfer, Research and Teaching Excellent Students (IPOKRaTES), International Union of Basic and Clinical Pharmacology (IUPHAR), clinical and translational section, and clinical pediatric pharmacology) were contacted and asked about formularies they are involved in or are aware of; (c) key persons identified for a specific neonatal formulary were asked also if they were aware of any other existing neonatal formulary. Using this strategy, we were also informed on alternative DI sources commonly used (such as textbooks), or the absence of national and supra-regional formularies in given countries or regions. It should be noted that the search for neonatal formularies was focused on independent and comprehensive drug databases; therefore, textbooks that contain neonatal drug information but are not dedicated to it (e.g., Yaffe and Aranda’s *Neonatal and Pediatric Pharmacology: Therapeutic Principles in Practice*, Nelson’s *Pediatric Antimicrobial Therapy or Pediatric Injectable Drugs (The Teddy Bear Book))* as well as specific neonatal guidelines were not considered for this work.

### 2.2. Questionnaire

The same core group developed (September–October 2022) a questionnaire targeted to describe the practices within a given formulary, with the intention to capture (dis)similarities in practices. Consecutive versions were assessed on face validity by this core group, as well as by contact persons of the specific formularies identified. The latest version (version 3.2, 19102022) is provided in the Appendix A. Responses were subsequently used as material for a narrative text and tabulated where appropriate to facilitate comparison between formularies.

### 2.3. Extraction Tool

To visualize potential differences in type and granularity of information, we extracted DI as provided in the latest version of the formularies for the top 10 most commonly administered drugs in neonates as defined by Stark et al. [11] To do so, we developed an extraction tool within the core group in consecutive steps and applied face validity. The latest version of this extraction tool is also provided in the Appendix A.

## 3. Results

### 3.1. Search Strategy

Authors/editorial board members of the following formularies were contacted: Australasian Neonatal Medicines Formulary (ANMF), British National Formulary for Children (BNF for Children), Dutch Pediatric Formulary (DPF, also known as ‘Kinderformularium’), Pediatric and Neonatal Lexi-Drugs (Lexicomp, by Wolters Kluwer^®^), NeoFax (available through Merative), SwissPedDose and Neonatal Formulary (Oxford University Press). Furthermore, colleagues from Canada, China and Japan shared information about neonatal DI availability in their respective countries, and an internet search led to an online Saudi handbook titled *Neonatal Dosage and Practical Guidelines Handbook 2nd Edition* whose author was contacted as well. Table 1 summarizes the descriptive characteristics of the abovementioned formularies and Figure 1 depicts the geographic points of origin for each formulary/colleague contacted.

### 3.2. Responses to the Questionnaire

In total, responses were received from representatives of six neonatal formularies: the Australasian Neonatal Medicines Formulary (ANMF), the Dutch Pediatric Formulary (DPF) and its international affiliates, Pediatric and Neonatal Lexi-Drugs (Lexicomp), NeoFax (Micromedex), the British National Formulary for Children (BNF for Children) and Neonatal Formulary (NNF). Key persons in the field of neonatal pharmacology from Canada, China and Japan were also contacted.

All of these formularies can be considered secondary sources of information, as all of them distill and aggregate data and recommendations directly from primary sources, such as published papers and reviews dealing with the neonatal population. In contrast, there are differences in aspects related to development and support, workforce and -flow, publication and update, and presentation of monographs. A brief description of each formulary is hereby given, based on the authors’ responses to the questionnaire. More information about the abovementioned key aspects for each of the formularies can be found in the Appendix A [12,13,14].

#### 3.2.1. Australasian Neonatal Medicines Formulary (ANMF)

The Australasian Neonatal Medicines Formulary (ANMF, previously known as Neomed) is an online formulary led by New South Wales (NSW) clinicians consisting of newborn specialists, pharmacists, nurses and other subspecialist groups, including infectious disease specialists, cardiologists, endocrinologists, gastroenterologists and neurologists, who are all from tertiary neonatal intensive care units. ANMF group also has international collaborators who are experts in neonatal pharmacology (KA, TY) and provide regular expert advice on all monographs before their release. ANMF standardizes medicinal treatment for (pre)term neonates by developing evidence-based consensus clinical guidelines for drugs used in this population. The project was commenced in late 2013 and, as of December 2022, over 150 monographs relevant to the neonatal population have been developed. On average, 40 meetings are held every year to stay agile and dynamic and to keep the content up to date. In addition to digital versions publicly available online, printable versions in pdf formats are circulated by ANMF to all tertiary NICUs and major special care nurseries within Australia and New Zealand.

The project is supported by the Clinical Excellence Commission, the NSW Ministry of Health and the Australian and New Zealand Neonatal Network (ANZNN). ANMF monographs are currently the only neonatal drug formulary endorsed by the NSW Ministry of Health and incorporated into electronic medical records in all NICUs statewide. ANMF steering group conducts weekly meetings using a structured agenda. In these 2 h online meetings, new as well as existing monographs that require review and updating based on feedback from hospitals are discussed. ANMF does not receive sponsorship from any commercial or pharmaceutical industry, nor does it use advertising from any source.

#### 3.2.2. British National Formulary for Children (BNF for Children)

British National Formulary for Children (BNF for Children) was created by the British National Formulary (BNF) in 2005 (BNF was first published in 1949 and served as a joint adults and children drug formulary until the emergence of BNF for Children). BNF for Children was developed originally by collaboration between BNF and Medicines for Children (MfC), an earlier British pediatric formulary first published in 1999 by RCPCH Publications Ltd. BNF for Children combined the knowledge gathered in MfC by British neonatal and pediatric experts from the Royal College of Paediatrics and Child Health (RCPCH) and the Neonatal and Paediatric Pharmacists Group (NPPG) with the expertise of BNF in publishing definitive drug formularies [15]. BNF and BNF for Children are funded from sales made by the joint publishers, namely, the BMJ Publishing Group and the Pharmaceutical Press, and RCPCH Publications Ltd. BNF for Children is available in print (published annually) and as an online web-based platform, the latter being available either independently (for UK-based users) or as a part of MedicinesComplete^®^ suite (for worldwide subscribers). It is also available as a mobile app in the UK.

#### 3.2.3. The ‘Dutch’ Pediatric Formulary (DPF, Also Known as ‘Kinderformularium’)

The ‘Dutch’ Pediatric Formulary (DPF) is a freely-available, government-financed, stand-alone online pediatric and neonatal formulary, whose target audience is all healthcare professionals providing, prescribing and administering drugs to children, i.e., physicians, pharmacists, nurses and pharmacy technicians. It was developed to provide dosing guidelines and accompanying DI based on the best available evidence from registration data, investigator-initiated research, relevant professional guidelines, clinical experience and consensus [16].

DPF is also available in country-specific versions in Germany, Austria and Norway with teams in each country adding country-specific information on licensing, availability of formulations and selecting those drugs, indications and age groups that are relevant to the clinical needs in the respective countries. The core pediatric and neonatal DI, however, is synchronized for all countries.

#### 3.2.4. NeoFax

NeoFax is a U.S.-based neonatal formulary, first published as a printed book in 1987 and updated annually. Thomson Corporation acquired NeoFax in 2007 and discontinued the print edition in 2011, opting for digital delivery exclusively to increase access and frequency of updates. Prior to acquisition, several localized editions were produced, e.g., a German edition titled *NeoFax: Arzneimittelhandbuch für die Neonatologie* (Gedon & Reuss, 2003), a Brazilian edition titled *NeoFax 2006: guia de medicações usadas em recém-nascidos* (São Paulo: Atheneu, 2006) and a Polish edition titled NeoFax^®^: *leki w neonatologii* (Ośrodek Wydawnictw Naukowych, 2006), among others, but none are still maintained and printed. NeoFax is currently available as an online web-based solution and a native mobile application, both in English only. The content includes neonatal-focused drug monographs, infant formula and human milk fortifier nutritional information (only for formulations available in the United States), as well as age- and indication-specific drug dosing calculators.

NeoFax’s target audience is healthcare professionals working in NICU settings, who may access NeoFax as a stand-alone solution or as part of a larger decision support system called Micromedex^®^ offered by Merative. Additional non-neonatal-specific tools that are available through Micromedex complement NeoFax, e.g., a drug interaction module, an IV compatibility screening tool, toxicology and reproductive effects content and pediatric DI (a module similar to NeoFax but focusing on infants and children).

#### 3.2.5. Neonatal Formulary (NNF)

*Neonatal Formulary—Drug Use in Pregnancy and the First Year of Life* is a British formulary in the form of a printed book and an online digital version, currently authored by Dr. Sean Ainsworth, and published by Oxford University Press. The NNF began in 1978 as a loose-leaf A4 reference folder at the Hospital for Sick Children in Newcastle upon Tyne. Written by Dr. John Inkster (Consultant Paediatric Anaesthetist) and Dr. Edmund Hey (Consultant Paediatrician), the formulary was updated regularly but retained the same format and basic layout. Until 1989, monographs reflected practice in the two neonatal units in Newcastle but then began to draw on accumulated experience throughout the Northern Regional Health Authority when it became known as the ‘Northern Neonatal Pharmacopeia’. Small pocketbook versions were published and used in neonatal units in the Health Authority in 1991 and 1993. In 1996, the ninth edition of this pharmacopeia found a national UK-wide audience when it was published by the BMJ Publishing Group as the first edition of the ‘Neonatal Formulary’. Monographs then began to reflect wider use across the UK and other countries. In 2004, publishing moved to Blackwell Publishing before moving to Oxford University Press in 2020.

This formulary offers both prenatal and neonatal DI to provide a narrative of drug effects from fetal to neonatal state. Thus, unlike most neonatal formularies, NNF also offers advice across every critical stage in the life of a neonate, starting with in utero exposure (placental transfer and teratogenicity), through safety in breastfeeding, to postnatal pharmacotherapy. It may be worth noting here that in many countries, DI related to pregnancy and lactation is provided by dedicated teratology information services, and therefore, it is not included in most neonatal formularies.

#### 3.2.6. Pediatric and Neonatal Lexi-Drugs (Lexicomp)

Lexicomp is a comprehensive DI database powered by Wolters Kluwer^®^, a global information services company, containing both adult and pediatric/neonatal drug monographs (the latter may be found under the ‘Pediatric and Neonatal Lexi-Drugs’ module) and other specific DI modules (e.g., Trissel’s IV Compatibility, Interactions, Drug I.D., Patient Education and more). *Lexicomp Pediatric & Neonatal Dosage Handbook* was first published in print in 1992 and continues to this day (November 2022 saw the 29th edition in print), while in 2005 the content became available online as a web-based platform. Lexicomp is currently available as a web-based platform, a mobile application and in print. While DI itself is available in English, the online user interface is available in 18 languages.

Neonatal DI is part of the pediatric database and includes a wealth of information, such as dosing information, pregnancy and lactation information (including Briggs’ Drugs in Pregnancy and Lactation, which provides a separate monograph for pregnancy and lactation information on top of information available in Lexicomp), usual IV concentrations, drug interactions, IV compatibility and more. It is designed to support all types of neonatal healthcare professionals, i.e., physicians, nurses, pharmacists, students and others, in inpatient, outpatient and retail pharmacy settings with details on local preparations in a variety of countries and regions around the world. Lexicomp is subscription-dependent, available to individuals and institutions, either as a standalone platform or as part of other information platforms, such as Wolters Kluwer’s UpToDate. It should be noted that certain materials are made widely available for free, usually in response to global healthcare emergencies (e.g., COVID-19).

### 3.3. Additional Information on Neonatal Formularies in Other Regions or Countries

At present, no national neonatal formulary is available in Canada, China or Japan. However, key persons in the field of neonatal pharmacology from those countries have provided guiding information on the availability of neonatal DI in these specific countries.

#### 3.3.1. Canada

Efforts have been made (and are still ongoing, to the best of our knowledge) to establish a Canadian pediatric and neonatal formulary. However, these intentions have not yet taken form, so no uniform neonatal formulary specific to Canada is currently available. It should be noted, however, that Pediatric and Neonatal Lexi-Drugs (Lexicomp) is known to be widely used in Canada and that it also contains monographs for drugs available only in Canada.

#### 3.3.2. China

The Chinese National Formulary (CNF) serves as China’s official drug formulary. The first edition of CNF was published in 2010 and did not distinguish between adult and pediatric DI. In 2013, however, a separate pediatric formulary was published, titled *Chinese National Formulary for Children (CNFC)*. It was revised for the first time in 2021.

CNFC writing and publishing are supported by the Chinese National Ministry of Health. It is published as a printed book in Chinese, meant to be used by healthcare professionals. There is no separate section for neonates. Rather, neonatal DI is available alongside DI for older children, captured under a dedicated subheading within the monograph. Monographs are based on evidence from the literature and expert consensus where literature is lacking. No specific information on neonatal, pediatric PK/PD or pediatric product labeling is available.

#### 3.3.3. Japan

No single dedicated neonatal formulary is available in Japan. Several printed books are used as references for neonatologists, and electronic medical records systems used in NICUs in Japan also provide dosing references to the user, as well as dosing alerts. In general, this DI is based on experience and consensus, and not necessarily on evidence from medical literature. The list of drugs for which neonatal DI is available includes drugs commonly used in NICUs and is expanded according to neonatologists’ requests. Information in these references is not updated regularly, update frequency ranges between 5–10 years and usually consists of minor updates as they become available. Extrapolations from PK of older children are rarely applied, but only for specific drug classes, such as antibiotics.

### 3.4. Comparison of Neonatal Formularies

To complement the above narratives of the six neonatal formularies reviewed here—NeoFax, ANMF, DPF, NNF (print version, 8th edition), Pediatric and Neonatal Lexi-Drugs and BNF for Children (online version, through Medicines Complete)—a side-by-side comparison of the formularies is presented in Table 2, Table 3, Table 4, Table 5, Table 6 and Table 7 which compare general characteristics, organization, dosing recommendations, preparation, and administration and monitoring data, respectively. This comparison lists the main aspects of the framework and organization of different drug formularies in general, such as the number of available monographs (at the time of writing), the listing of references, the type of funding, etc., as well as aspects that are unique to neonatal drug formularies, such as differentiation in DI between term and preterm neonates, neonatal adverse effects, specific monitoring parameters, etc. These tables have been constructed based on data provided by formularies’ authors and data gathered from each formulary using a structured data extraction tool Appendix A for the 10 most commonly used drugs in NICUs in the United States, according to a recent paper by Stark et al. (2022) [11]: ampicillin, gentamicin, caffeine citrate, poractant alfa, morphine, vancomycin, furosemide, fentanyl, midazolam and acetaminophen. The full data extracted for these top 10 drugs can be found in the Appendix A.

To further illustrate differences and similarities between the examined neonatal formularies, the dosing sections of three common drugs in the NICU—caffeine citrate, morphine and gentamicin—were compared, and dosing recommendations were subsequently extracted along with directly related comments (Table 8, Table 9 and Table 10 respectively). To avoid cluttering and excess information, only dosing recommendations for the main indication of each drug were extracted. An attempt was made to maintain original wording as much as possible on one hand, while formatting all dosing recommendations as uniformly as possible on the other.

Out of the six examined neonatal formularies, three have a specific focus on neonatology neonatal-specific (ANMF, NeoFax and NNF), whereas the other three provide pediatric DI as well. Pediatric and Neonatal Lexi-Drugs has the largest number of monographs overall (1460), followed by BNF for Children (1031), DPF (884), NeoFax (304, neonatal only), NNF (265) and ANMF (182). The number of neonatal-specific monographs was not obtainable from all formularies, but they range between slightly less than 200 and over 350 neonatal-specific monographs.

Almost all formularies provide specific references for DI, both in-line and collectively in a designated section of the monograph. References are usually from the primary literature and/or product information (SmPC). Only BNF for Children does not provide references but does offer a direct link to the British SmPC in the Electronic Medicines Compendium (emc), on which at least some of the DI is based (of note, the emc is a freely available resource for DI on medicines licensed for use in the UK, unrelated to BNF for Children). All formularies are peer-reviewed as part of their validation process, and all are independent of the pharmaceutical industry. Commercial products are listed in all formularies, with each formulary listing some or all of the commercial products available in the country/region it targets.

With regard to the organization of information, all formularies differentiate between preterm and term infants, and use conventional and accepted neonatal terminology to differentiate between neonatal subpopulations, as age and birthweight are usually a good indication of the level of maturation of the patient, from which physiological status (e.g., hepatic and renal function, lung and gastrointestinal maturity, etc.)—which affects PK parameters, and therefore the expected level of response to drug therapy—is derived.

Additionally, all formularies denote specifically the route of administration appropriate for each dosing recommendation, and all provide hepatic and renal dosage adjustments as appropriate, as well as other types of dosage adjustments required by altered physiological status (e.g., therapeutic hypothermia and ECMO).

Regarding the drug preparation aspect, all formularies besides DPF offer specific information and guidance on how to best prepare the drug, ranging from manufacturer’s instructions to extemporaneous preparations derived from the primary literature. However, some formularies provide more data than others regarding preparation information. Thus, as mentioned, DPF does not provide preparation data at all, while ANMF monographs published since 2020 do provide optimal and maximal concentrations where applicable (notably, no such information was identified for the 10 studied drugs Appendix A), and NNF sometimes provides optimal concentrations but no maximal concentrations for any of the 10 drugs studied.

All formularies provide information on the method of administration, i.e., duration of infusion, distance from feeding, etc., and all formularies provide monitoring advice, usually in a dedicated section, which details monitoring parameters, suggested timetables and desired plasma levels where appropriate.

All formularies provide drug interactions information and adverse effects in dedicated sections (NNF monographs are inconsistent regarding the inclusion of such section), save for NeoFax, which does not include a drug interactions section within each monograph (a drug interactions module is available separately to Micromedex subscribers). All formularies save for ANMF also provide frequencies of adverse effects where available, either in percentages or categorically.

All formularies mention notable excipients with potential effects on neonates. PK data is also provided in all formularies, although not in the same manner. NeoFax, ANMF and NNF provide only neonatal PK data where available, DPF provides PK data in older children, if available, as a comparison for neonatal data, and Pediatric and Neonatal Lexi-Drugs provides neonatal and pediatric PK data where available, and adult PK data is provided where unavailable.

Dosing recommendations for caffeine, morphine and gentamicin illustrate some similarities and differences between the formularies.

Caffeine usually appears in commercial products as caffeine citrate. All formularies indicate dosing both for caffeine citrate and caffeine base as well as the proportion between them (2:1), and always clearly indicate if the recommended dosing is for the citrate salt or the base compound. Regarding the dosing recommendations themselves, they are fairly consistent across formularies: all formularies mention the need for a loading dose (LD), usually 20 mg/kg of caffeine citrate, although some formularies (NeoFax and NNF) allow for LD of up to 25 mg/kg. Additionally, both Pediatric and Neonatal Lexi-Drugs and NNF provide information for even higher LD (up to 80 mg/kg), and Pediatric and Neonatal Lexi-Drugs states that such dosage should be used with caution, detailing adverse effects observed under such dosage, while NNF mentions that such LD is not for treatment of apnea of prematurity but for facilitating extubation in neonates born under 30 weeks of gestation. Maintenance doses (MD) range between 5 and 10 mg/kg/day of caffeine citrate across formularies. ANMF, BNF for Children, NeoFax and NNF also recommend adjusting MD based on clinical need. All formularies provide both routes of administration (IV and oral), while BNF for Children, DPF and NNF state explicitly that the injection solution may be administered orally (oral preparations of caffeine, either commercial or extemporaneous, are widely available). Only Pediatric and Neonatal Lexi-Drugs and NeoFax address premature neonates explicitly with specific age-based dosage recommendations.

Morphine may be given to neonatal patients through several routes of administration. All formularies indicate the oral and IV route, which are the most common routes. However, subcutaneous injection is also possible according to BNF for Children, Pediatric and Neonatal Lexi-Drugs, NeoFax and NNF, whereas IM injection is mentioned only by Pediatric and Neonatal Lexi-Drugs and NeoFax, and dosing recommendations for the rectal route are only available on DPF. All formularies suggest IV dosing via a bolus LD followed by continuous infusion MD with similar dosage recommendations, with all but ANMF explicitly stating that doses should be individualized/given as needed.

Dosing recommendations vary according to the route of administration but are overall fairly comparable across formularies. However, units of measurement differ between formularies: ANMF, BNF for Children and NNF report dosage recommendations in mcg/kg, whereas DPF, Neonatal Lexi-Drugs and NeoFax report their dosage recommendations in mg/kg. While the dosages are essentially the same, the fact that different formularies choose to express units of measurement differently may sometimes lead to dosage and/or calculation errors, depending on the habits of the clinician, as well as the units commonly used in their institution. Furthermore, specific, age-based recommendations for preterm infants are available on ANMF, DPF and Pediatric and Neonatal Lexi-Drugs.

Gentamicin is a commonly used antibiotic in neonatal units, mainly for its efficacy in the empiric treatment of suspected neonatal sepsis. It is usually given through the IV route, although IM injections are also possible (albeit less preferred). All formularies except DPF state that IM injections are possible (with reservations). It should be noted that BNF for Children and NeoFax do not provide this information in the neonatal dosing section but in other sections of the monograph.

Dosing recommendations for gentamicin are largely similar across formularies, as are the dosing intervals, which are affected by the age of the infant since it has a rather good correlation with the degree of renal function because gentamicin is eliminated mostly renally. The age categories for determining optimal dosing interval slightly differ between formularies, with NeoFax and Pediatric and Neonatal Lexi-Drugs suggesting six age categories and, on the other end, BNF for Children suggests only two age groups. All formularies besides BNF for Children address in the neonatal dosing section the need for therapeutic drug monitoring during treatment with gentamicin. BNF for Children does address this issue, not in the neonatal dosing section but in the ‘monitoring parameters’ section.

## 4. Discussion

With this project, we intended to provide a robust snapshot of the different neonatal formularies available. To the very best of our knowledge, this is the first attempt to collect information on the function and approaches taken by the leading neonatal formularies. Only one somewhat similar attempt was encountered in the literature, in which Zenk listed pediatric texts and journals and classified them as texts that should be included in a basic pharmacy library, or more ‘advanced’ pediatric texts [17]. However, this work focused on textbooks and journals and not actual formularies; it was not dedicated to the neonatal population and did not compare these resources in depth. This is of relevance, as it became clear through this work that there are both similarities and dissimilarities in workflow, organization and type of information provided, the knowledge of which may help to direct clinicians to the most appropriate formulary for specific DI. Therefore, it was quite interesting to notice that this type of information is not easily available in the public domain, although DPF and Swisspeddose have published their workflow and approach, whereas ANMF describes their workflow and governance on their website [16,18,19]. Other formularies provide information on how to use their platform and how it is structured in their own help sections, which are available to subscribers only.

Our second objective for this paper was to increase the awareness of clinicians of all healthcare professions taking care of neonates, in particular those who are relatively new to this specialty, to as many neonatal formularies as possible. As most neonatal formularies were established as local initiatives with (mostly) local readership in mind, it may be that healthcare professionals who are aware of a ‘native’ formulary available to them may be unaware of other formularies which may be also accessible to them and provide them with DI that their native formulary may not provide. Even though ‘native’ formularies aim to be finetuned to their target audience, periodic changes in drug marketing and distribution, new drug authorizations and changes in drug requirements of the population may require formularies to rapidly adapt to such changes, which may prove challenging, given the rigorous processes all formularies employ for monograph development and updating. This gap may be bridged easily by using other neonatal formularies which may already provide relevant DI. Moreover, healthcare professionals with no ‘native’ formulary may tend to use whatever formulary was made available by their institution, often without a thorough examination or even outright awareness of other available formularies. Such practice may cause those neonatal professionals to be unaware of other potentially useful formularies, thus limiting the amount of DI available for making clinical decisions. Given the small number of identified formularies and the seemingly random geographical distribution (stemming from the formularies being local initiatives rather than a globally organized effort, as mentioned), we felt that a global scope review would provide a most detailed picture of the current state of neonatal formularies.

We observed different business models, which can be divided as follows: (a) models supported by public resources or by the learned professional societies or similar (ANMF, DPF and Swisspeddose); (b) private commercial initiatives, commonly incorporated in information services companies (NeoFax, Pediatric and Neonatal Lexi-Drugs and BNF for Children); or (c) scientific publishers (NNF). It should be noted that government funding may in certain cases allow making a formulary freely and publicly available. This aspect should not be underestimated, as having a freely available credible DI source grants healthcare professionals with limited resources and funds (e.g., in developing countries) access to validated DI, which would otherwise be unavailable to them. Related to the workforce, there is a diversity of profiles involved, e.g., neonatologists, clinical pharmacologists, neonatal and/or hospital pharmacists as well as some diversity in its organization. As reflected in Section 3.2, there is more diversity in workflow, commonly based on a system with monograph development (structured search strategies) and subsequent internal or external review and assessment. Along the same procedures, all formularies have processes related to the communication of updates, but with differences in practices and timing. We should not underestimate the relevance of such updates, as timely and accurate updates are very likely the most effective approach to truly implement the best-evidenced DI into clinical practice. Changes in dosing recommendations, newly observed adverse effects and new data on the stability of preparations have a direct influence on the efficacy and safety of neonatal pharmacotherapy. Formularies hereby hold the promise to overcome the well-known information and implementation gap [6]. Being secondary sources of information, formularies are relied on to be accurate and up-to-date by users, as they save clinicians precious time in researching primary literature. If formularies maintain open communication with their users through an active updating system (e.g., in-system pop-up messages and periodic newsletters), this will contribute to clinicians’ confidence in their practice.

Monographs in the individual formularies report along a predefined template for each formulary, but differ (see Appendix A) in the granularity of information, such as renal or hepatic dosage adjustments, drug–drug interactions, most relevant adverse effects, or preparation (optimal concentration, maximal concentration and excipients) and stability data.

Even so, a bird’s-eye view of all formularies reveals a high level of similarity between them. This finding is reasonable for the following two reasons: (a) clinical parameters and DI requirements are not expected to be different across different regions of the world; (b) the literature on neonatal pharmacotherapy is relatively limited, and thus it is probable that, to some extent, formularies will use the same sources as the basis for their DI. Differences in chosen references, however, may reflect on editorial preferences, which in turn may directly reflect on the provided DI. The process of making these editorial choices may be interesting in itself but is out of the scope of this paper.

Perhaps the most clinically useful aspect of DI is the dosing recommendations, as this is the information that eventually determines how the patient will receive the drug. In general, optimal dosing recommendations are determined by the manufacturer of the drug following phase II clinical trials, also known as ‘dose-finding’ studies [20], but these studies rarely include neonates, not to mention preterm neonates. Thus, neonatal dosing recommendations should be formulated cautiously and stem from established data. From the comparison conducted here between caffeine, morphine and gentamicin, the dosing recommendations appear to be highly similar, which again may be explained by the relatively limited data which is available to all formularies (although the similarities and differences in references have not been thoroughly examined here). However, there are differences between dosing recommendations as well, and these differences pose the greatest question to the clinician: which dosing recommendations should be chosen? How should one decide between different dosing recommendations appearing in different formularies? 

The decision can rely on the number and type of references; i.e., how much data is available in each formulary to support the dosing recommendation and what is it based on. For example, some dosing recommendations may be based on a case report or case series reporting on a few patients or even a single neonate, whereas others may be based on an observational study with tens or hundreds of patients. In other words, the strength of the recommendation may depend on the strength of the evidence. 

The methodology of dose selection may also play a part here, i.e., how are dosing recommendations formulated? Are the doses reported from efficacy studies in neonates, or from pharmacokinetic/pharmacodynamic studies, or are they an integration of all available information, supplemented by expert opinions?

Another factor that should be taken into consideration would be consensus among formularies. If several formularies state a certain dosing recommendation and another one states a different dosing recommendation, perhaps it would be best to follow the majority (unless all depend on the same source, in which case consensus plays no role in the uniformity of recommendation). A fourth consideration would be the clinical judgment of the prescriber. The clinician may prescribe a dose at the high end of the dosing range if a strong and/or rapid response is required and the patient is deemed stable enough, whereas a lower dose may be preferred for complicated and/or more fragile patients. The expertise and conservativeness of prescribers may also play a part in the decision.

In addition to dosing recommendations and related information, such as administration and monitoring, which form the basis for neonatal DI, the examined formularies do not follow a uniform pattern. Broader pharmacology aspects are also considered in all formularies, although the type of pharmacological information differs between formularies. Information on drug preparation was found to be absent from some formularies, either partially or entirely, making certain formularies useful as guides for the practical preparation of the drug, whereas others only provide ‘theoretical’ DI.

We believe that prescribers should be aware of this variability in the granularity and type of information, as failure to retrieve information on, e.g., optimal concentrations, solution compatibility or stability in a given formulary does not necessarily mean that this type of information does not exist (although sometimes it might be the case), as it might be available elsewhere. To put it in perspective, ANMF, Pediatric and Neonatal Lexi-Drugs, NeoFax and NNF may be regarded as comprehensive sources for neonatal DI, covering both pharmacology and practical aspects of preparation as well as commercial product listing. A caveat for NeoFax is that it does not provide in-monograph information on drug–drug interactions—it should be borne in mind that this information is available separately in Micromedex. DPF and BNF for Children were found to be mainly concerned with pharmacology and safety of the drugs they provide information for, with no or little regard to preparation information. As elucidated in this work, the choice of which information to include in a formulary is at the sole discretion of the author(s). Thus, one possible explanation is that those formularies do not address preparation information due to the availability of this data from other sources in common use in their respective regions. Naturally, the variety of drug monographs may also differ between formularies, in accordance with drug availability in the origin and/or target region of each formulary (e.g., ampicillin, which is not available in the Dutch market).

Another point that should be taken into consideration is the specificity of formularies to neonatal DI and whether it has any influence on the amount and type of neonatal DI. As demonstrated, some formularies have a specific focus on preterm and term neonates (ANMF, NeoFax and NNF), whereas others (BNF for Children, DPF, Pediatric and Neonatal Lexi-Drugs and others, see Table 1) offer both pediatric and premature and term neonatal DI in their drug monographs. Although not directly examined, it seems that the choice of formulary scope in itself does not reflect on the amount or type of DI presented, or its level of detail. These aspects probably depend on the size of the writing/editorial team and the time dedicated to writing new monographs and revising existing ones, which naturally differ between formularies. Thus, users should not depend on formulary specificity for a certain population or number of monographs in deciding whether or not it may fit their needs. Most formularies may be able to offer a limited-time trial period, which would allow potential users to see the quality of the formulary for themselves.

It should be mentioned that reasonable efforts were made to contact all neonatal formularies known to us or identified in our explorative search. However, not all contact attempts were successful, likely because we failed to identify relevant key contacts for our specific requests. Thus, this review constitutes our best effort to compose a review as comprehensive as possible, but we acknowledge the fact that the picture is not complete, albeit rather detailed.

To summarize, robust, evidence-based neonatal DI is crucial for neonatal clinicians, as drug labels often provide little to no relevant information for this population. Therefore, neonatal formularies are often the only information source that collects and processes neonatal pharmacology literature to produce clinically useful DI and to facilitate decision making in NICU settings. Neonatal clinicians should be familiar with as many neonatal formularies as possible, to make sure their needs are met when it comes to neonatal pharmacology questions and problem-solving. Hopefully, this review will allow neonatal professionals and institutions to make informed decisions on which formulary(ies) they should consult and have on hand, making neonatal DI sufficiently accessible to ensure the highest efficacy and safety of neonatal pharmacotherapy.

## 5. Conclusions

In an effort to identify and describe the available neonatal formularies, eight different neonatal formularies were identified worldwide (Europe, USA, Australia-New Zealand and Middle East). Six of them responded to the questionnaire and were compared for structure and content. Each formulary has its own workflow, monograph template and style, and update routine. Focus on certain aspects of DI also varies, as well as the type of initiative and funding. Clinicians should be aware of the various formularies available and their differences in characteristics and content to use them properly for the benefit of their patients.

## Figures and Tables

**Figure 1 children-10-00848-f001:**
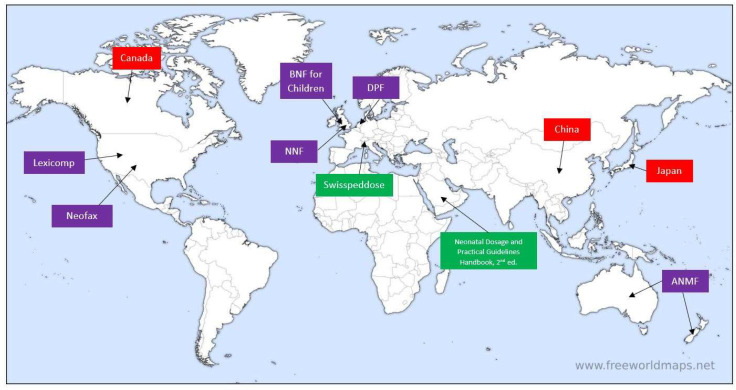
Geographic pinpointing of contacted formularies/colleagues. Purple boxes indicate contacted formularies that responded; boxes in green indicate contacted formularies that did not respond; boxes in red indicate contacted colleagues where no formularies were found. ANMF = Australasian Neonatal Medicines Formulary, BNF for Children = British National Formulary for Children, DPF = Dutch Pediatric Formulary, NNF = Neonatal Formulary. (Image source: www.freeworldmaps.net, used with permission in agreement with source’s terms and conditions of use).

**Table 1 children-10-00848-t001:** Descriptive characteristics of included neonatal formularies.

#	Name	Publisher	Country	Language	Format	Access	Edition, a Year of Printed Publication	URL/ISBN
1	Australasian Neonatal Medicines Formulary (ANMF)	Australasian Neonatal Medicines Formulary (ANMF) consensus group	Australia and New Zealand	English	Digital and print	Public		https://www.anmfonline.org/, (accessed on 1 May 2023)
2	*British National Formulary for Children (BNF for Children)*	BMJ, Pharmaceutical Press and RCPCH Publications Ltd.	United Kingdom	English	Digital and Print	Public (UK) Subscription (elsewhere)	2022–2023	9780857114297 https://bnfc.nice.org.uk/ (UK only access for the UK health service)(not accessed, as UK Health restricted) Worldwide access is available through a subscription to MedicinesComplete: https://about.medicinescomplete.com/ (accessed on 1 May 2023)
3	Dutch/German/Austrian and Norwegian Pediatric Formularies (DPF, also known as ‘Kinderformularium’)	Foundation Dutch Knowledge Center Pharmacotherapy for Children and Kinderformularium BV	The Netherlands, Together with Norway, Germany and Austria	Dutch German Norwegian	Digital	Public		https://www.kinderformularium.nl/ www.kinderformularium.de www.koble.info www.kindermedika.at (accessed on 1 May 2023)
4	Pediatric and Neonatal Lexi-Drugs (Lexicomp)	Wolters Kluwer^®^	USA	English (online interface is available in 18 languages)	Digital (web-based and mobile application) and Print	Subscription	2022–2023; 29th Edition	9781591953913 https://online.lexi.com/lco/action/home (accessed on 1 May 2023)
5	NeoFax	Merative	USA	English	Digital (web-based and mobile application)	Subscription		https://www.micromedexsolutions.com/home/dispatch/ (accessed on 1 May 2023)
6	*Neonatal Formulary*	Oxford University Press	United Kingdom	English	Digital and Print	Full online version requires subscription	8th, 2020	9780198840787 Supplementary material for the print version is freely available at: https://academic.oup.com/book/35484#login-purchase (accessed on 1 May 2023)
7	*Neonatal Dosage and Practical Guidelines Handbook*	Saudi Ministry of Health	Saudi Arabia	English	Print		2nd, 2016	9786038144848 (accessed 1 May 2023) Freely available at the Saudi Ministry of Health website: https://www.moh.gov.sa/HealthAwareness/EducationalContent/BabyHealth/Documents/book.pdf (accessed on 1 May 2023)
8	SwissPedDose	Swiss Society of Neonatology	Switzerland	German, French, English, Italian	Digital	Public		https://swisspeddose.ch/ (accessed on 1 May 2023)

**Table 2 children-10-00848-t002:** Content and organization comparison between neonatal formularies: General characteristics.

Parameter	Neofax	ANMF	DPF	NNF	Pediatric and Neonatal Lexi-Drugs	BNF for Children
Pediatric and Neonatal formulary or neonatal only	Neonatal only (a parallel module is available for pediatric patients)	Neonatal	Neonatal and pediatric	Neonatal and pediatric (infancy)	Neonatal and pediatric	Neonatal and pediatric
Number of drug monographs included overall	304 (pediatric: 804)	182	884	265 (covering over 300 unique drugs or drug combinations)	1460 (376 with neonatal dosing)	1031
Number of monographs with DI for term neonates	304	182	252	N/A	360	218
Number of monographs with DI for preterm neonates	112	182	73	N/A	336	N/A
Are references and citations included?	Yes, in-line and collectively under a ‘References’ section	Yes, in-line and collectively under a ‘References’ section	Yes, in-line and collectively under a ‘References’ section	Yes, collectively in the end of the monograph	Yes, in-line and collectively under a ‘References’ section	No
Is communication between users and authors possible?	Yes	Yes	Yes, contact form “suggestions for improvement” is included in each monograph, delivered to editorial team	Yes (through email to publishers)	Yes, via online customer support	Yes, clinical content enquiries can be made via email
Peer-reviewed	Yes	Yes	Yes	Yes	Yes	Yes
Funded by public/not for profit or private/commercial	Commercial	Public/not for profit	Public/not for profit	Commercial	Commercial	Commercial
Independent from pharmaceutical industry	Yes	Yes	Yes	Yes	Yes	Yes
Are commercial products listed?	Yes (available formulations and strengths)	Yes	Yes (available formulations and strengths are listed)	Yes	Yes, in Brand Names sections for US, Canada and International (where appropriate)	Yes

N/A = Not available.

**Table 3 children-10-00848-t003:** Content and organization comparison between neonatal formularies: Organization.

Parameter	Neofax	ANMF	DPF	NNF	Pediatric and Neonatal Lexi-Drugs	BNF for Children
Differentiation between preterm and term patients?	Yes, usually under the ‘Dose’ section	Yes	Yes	Yes	Yes, usually under the ‘Dosing: Neonatal’ section	Yes
Categories used for preterm age groups (birthweight, PNA, PMA, current weight)	Yes, based on the evidence; usually under the ‘Dose’ section	Where appropriate and indicated	If available from literature: birthweight, PNA, PMA and PCA	Where appropriate and indicated	Where appropriate and indicated; under the ‘Dosing: Neonatal’ section	Where appropriate. Usually based on weight. Corrected gestational age
Dosing recommendations are differentiated by routes of administration?	Yes	Yes	Yes	Yes	Yes, every dosing recommendation is associated with route(s) of administration	Yes
Transparency on how information is compiled	Yes	Yes	Yes	Yes	Yes	Yes
Information can be verified by references/source documentation	Yes	Yes	Yes	Yes	Yes	Yes
Extrapolation from older pediatric age groups or adults based on available data	No (would accept published literature with extrapolated data)	Yes	Yes	N/A	Evidence for dosing provided with inline citations	Yes

PMA = post-menstrual age, PNA = post-natal age, PCA = post-conceptional age, N/A = Not available.

**Table 4 children-10-00848-t004:** Content and organization comparison between neonatal formularies: Dosing recommendations.

Parameter	Neofax	ANMF	DPF	NNF	Pediatric and Neonatal Lexi-Drugs	BNF for Children
Dosing recommendations are differentiated by indication?	Yes	Yes	Yes	Yes	Yes, every dosing recommendation is associated with indication(s)	Yes
Renal dosage adjustment (if applicable)	Yes, under the ‘Dose’ section	Yes, since 2020	Yes, specific section, including advice on adjustment based on AKI for > 3 months of age. Dose adjustment for neonatal maturation included in neonatal dosing recommendation.	Yes	Yes, under the ‘Dosing: Neonatal’ section	Yes
Hepatic dosage adjustment (if applicable)	Yes, under the ‘Dose’ section	Yes, since 2020	If applicable and available under ‘Warnings and Precautions’ section.	Yes	Yes, under the ‘Dosing: Neonatal’ section	Yes
Other dosage adjustments (if applicable)	Yes, under the ‘Dose’ section (e.g., TH and ECMO)	Yes, since 2020	If available from literature (TDM, ECMO, Hypothermia, obesity and pharmacogenetics).	Yes	Yes, under ‘Dosing: Neonatal’, as Notes at the top	Yes

AKI = acute kidney Injury, TH = therapeutic hypothermia, ECMO = extracorporeal membrane oxygenation.

**Table 5 children-10-00848-t005:** Content and organization comparison between neonatal formularies: Preparation.

Parameter	Neofax	ANMF	DPF	NNF	Pediatric and Neonatal Lexi-Drugs	BNF for Children
Are preparation instructions available?	Yes, under the ‘Special Considerations/Preparation’ section	Yes	No	Yes	Yes, under the ‘Preparation for Administration: Pediatric’. There is an ‘Extemporaneous Preparations’ section for bulk compounding of oral preparations, plus access to Trissel’s IV Compatibility.	Yes
Are stability data provided (if applicable)?	Yes, under the ‘Special Considerations/Preparation’ section	Yes	No	Yes	Yes, in the ‘Storage/Stability’ section, plus access to Trissel’s IV Compatibility	Yes
Optimal concentration of preparation (if applicable)	Yes, under the ‘Special Considerations/Preparation’ section	Yes	No	Yes	Yes	Yes
Maximum concentration of preparation (if applicable)	Yes, under the ‘Special Considerations/Preparation’ section	Yes	No	No	Yes	Yes
Is method of administration indicated?	Yes, under the ‘Administration’ section	Yes	Yes, under the ‘Dosages’ section	Yes	Yes, under the ‘Administration: Pediatric’ section	Yes

**Table 6 children-10-00848-t006:** Content and organization comparison between neonatal formularies—Administration and monitoring.

Parameter	Neofax	ANMF	DPF	NNF	Pediatric and Neonatal Lexi-Drugs	BNF for Children
Are monitoring advice and parameters available?	Yes, under the ‘Monitoring’ section	Yes	Yes	Yes	Yes, under the ‘Monitoring Parameters’ and ‘Reference Range’ sections	Yes
If monitoring advice and parameters are indicated, are specific neonatal warnings and precautions mentioned (if applicable)?	Yes	Yes	Yes, if available from the literature or clinical expertise	Yes	Yes	Yes

**Table 7 children-10-00848-t007:** Content and organization comparison between neonatal formularies: Pharmacological data.

Parameter	Neofax	ANMF	DPF	NNF	Pediatric and Neonatal Lexi-Drugs	BNF for Children
Drug–drug interactions mentioned?	No; an external drug–drug interactions module is available to Micromedex subscribers	Yes	Yes	Yes	Yes, under the ‘Drug Interactions’ section and in a separate module	Yes
Adverse effects mentioned?	Yes, under the ‘Adverse Effects’ section; frequency is provided where available	Yes	Yes, frequency is provided where available	Yes, a dedicated section is included only in certain monographs	Yes, under the ‘Adverse Reactions’ section; frequency is provided where available	Yes, frequency is provided where available
If adverse effects are mentioned, are specific neonatal effects mentioned?	Yes	Yes	Yes, if available from the literature or clinical expertise	Yes	Yes	Yes
Are notable excipients mentioned?	Yes, if applicable, under the ‘Contraindications/precautions’ section	Yes, since 2020	Yes (if available from product info, including concentration)	Yes	Yes, under the ‘Dosage Forms: US’ and/or ‘Dosage Forms: Canada’ sections	Yes
Neonatal Pharmacokinetic data available?	Yes, under the ‘Pharmacology’ section	Yes, wherever available	Yes, if available from the literature	Yes	Yes, under the ‘Pharmacokinetics (Adult data unless noted)’ section	Yes, often under the monitoring section

**Table 8 children-10-00848-t008:** Comparison of dosing recommendations for caffeine citrate indication: Apnea of prematurity.

ANMF	BNF for Children	DPF	Pediatric and Neonatal Lexi-Drugs	NeoFax	NNF
Caffeine citrate: Route: IV, Oral LD: 20 mg/kg MD: 10 mg/kg (range 5−20 mg/kg) daily Post-op apnoea (single dose): 10 mg/kg Caffeine Base: Route: IV, Oral LD: 10 mg/kg MD: 5 mg/kg (range 2.5−10 mg/kg) daily Post-op apnoea (single dose): 5 mg/kg Comments: Maintenance dose may be increased or decreased as per the clinical need.	Caffeine citrate: Neonate: Route: IV, Oral LD: 20 mg/kg MD: 5 mg/kg once daily, started 24 h after the loading dose Comments: Dose equivalence and conversion: Caffeine citrate 2 mg ≡ caffeine base 1 mg MD above 20 mg/kg/day can be considered if therapeutic efficacy is not achieved	GA < 37 weeks Caffeine citrate: Route: IV, Oral LD: 20 mg/kg/day MD: 5–10 mg/kg/day Caffeine base: Route: IV, Oral LD: 10 mg/kg MD: 2.5–5 mg/kg/day Comments: The injection solution can be administered orally.	Manufacturer’s labeling: Preterm neonates: Route: IV LD: 20 mg/kg caffeine citrate as a one-time dose Route: IV, Oral MD: 5 mg/kg/dose caffeine citrate once daily, beginning 24 h after loading dose Alternate dosing: GA < 32 weeks: Route: IV LD: 20–40 mg/kg caffeine citrate Route: IV, Oral MD: 5–20 mg/kg/dose caffeine citrate once daily starting 24 h after the loading dose Comments: Loading doses as high as 80 mg/kg of caffeine citrate have been reported, but should be utilized with caution. Dose expressed as caffeine citrate; caffeine base is ½ the dose of the caffeine citrate	FDA-Approved Dosage: Route: IV, Oral LD: 20–25 mg/kg of caffeine citrate IV over 30 min or orally. Route: IV, Oral MD: 5–10 mg/kg/dose of caffeine citrate IV slow push or orally every 24 h Off-label Dosage: GA 22–28 weeks: Route: IV LD: Median, 20 mg/kg IV (range, 19–23 mg/kg) Route: IV, Oral MD: Median, 8 mg/kg/day (range 5–10 mg/kg/day) Comments: May consider an additional loading dose and higher maintenance doses if able to monitor serum concentrations.	Route: IV, Oral LD: 20–25 mg/kg of caffeine citrate IV or orally. MD: 5 mg/kg IV or orally once every 24 h Comments: As the baby matures it may be necessary to increase the dose of caffeine citrate by an additional 1 mg/kg every one to two weeks to ensure that the minimal caffeine concentrations (15–20 mg/L) that are recommended for treatment of apnoea or prematurity are maintained. Drug equivalence: There is 1 mg of caffeine *base* in 2 mg of caffeine *citrate*.

LD = loading dose, MD = maintenance dose, GA = gestational age, PNA = postnatal age, PMA = postmenstrual age (equal to GA + PNA).

**Table 9 children-10-00848-t009:** Comparison of dosing recommendations for morphine indication: Analgesia.

ANMF	BNF for Children	DPF	Pediatric and Neonatal Lexi-Drugs	NeoFax	NNF
Route: PO Starting dose: 50–200 mcg/kgevery 3–6 h. Route: IV bolus 50 mcg/kg (maximum recommended 100 mcg/kg) every 4 h. Route: continuous IV infusion Range: 5–40 mcg/kg/h: Ventilated infants or after surgery: PNA 0–7 days:Starting dose: 10 mcg/kg/h Range: 5–40 mcg/kg/h PNA 8–30 days:Starting dose: 15 mcg/kg/h Range: 5–40 mcg/kg/h PNA 31–90 days:Starting dose: 20 mcg/kg/h Range: 5–40 mcg/kg/h Comments:Infants after cardiovascular surgery may need a lower starting dose and be titrated to clinical response.	Route: SC Dose: Initially 100 mcg/kg every 6 h, adjusted according to response. Route: IV Dose: 50 mcg/kg every 6 h, adjusted according to response, dose to be administered over at least 5 min, alternatively (by intravenous injection) initially 50 mcg/kg, dose to be administered over at least 5 min, followed by (by continuous intravenous infusion) 5–20 mcg/kg/h, adjusted according to response.	Route: PO Term Neonates Dose: 0.3–0.6 mg/kg/day in 6 doses. Route: Rectal Dose: 0.6–1.2 mg/kg/day in 6 doses. Route: IV Premature neonates Gestational age < 37 weeks and term neonates: Starting dose: 50–100 mcg/kg/dose over 60 min MD: 3–20 mcg/kg/h, as continuous infusion. Administer under supervision. If the effect is insufficient, the hourly dose can be given as a bolus injection, increasing gradually.	Route: PO Dose: 0.05–0.1 mg/kg/dose every 4 to 8 h as needed. Route: IM, IV (preferred), SC: Initial: 0.05–0.1 mg/kg/dose every 4 to 8 h as needed Comment: use lowest effective dose and titrate carefully as clinically indicated. Route: continuous IV infusion: Preterm and term neonates: Initial: 0.01 mg/kg/h (10 mcg/kg/h); titrate carefully based on patient response, considering benefits and risk of additional opiate exposure Comments: Continuous IV infusion: use with caution if patient at risk of hypotension; some experts recommend not using in neonates with GA < 27 weeks.	Route: IV, IM, SC Dose: 0.05–0.2 mg/kg per dose. Repeat as required (usually every 4 h). Route: IV continuous infusion LD: 0.1 mg/kg MD: 0.01 mg/kg/h Comment: postoperatively may be increased further to 20 mcg/kg/h.	Short-term pain relief: Route: PO Dose: 200 mcg/kg Route: SC, IV (preferred) Dose: 100 mcg/kg Comments: Rapid intravenous administration does *not* cause hypotension but may cause respiratory depression. A further 50 mcg/kg/dose can usually be given after 6 h without making ventilator support necessary. Severe or sustained pain: Route: IV LD: 200 mcg/kg MD: 20 mcg/kg/h Comments: Provide ventilatory support. While this [dosage regimen] will usually control even severe pain in the first 2 months of life, providing a plasma morphine level of 120–160 ng/mL, treatment *has* to be individualized. Staff needs discretion to give further 20 mcg/kg bolus up to once every 4 h to control any ‘break-through’ pain.

LD = loading dose, MD = maintenance dose, GA = gestational age, PNA = postnatal age, PMA = postmenstrual age (equal to GA + PNA), mcg = microgram, ng = nanogram, SC = subcutaneous.

**Table 10 children-10-00848-t010:** Comparison of dosing recommendations for gentamicin indication: Gentamicin-susceptible infection and neonatal sepsis.

ANMF	BNF for Children	DPF	Pediatric and Neonatal Lexi-Drugs	NeoFax	NNF
Route: IV, IM (only if IV access is not available) Dose: 5 mg/kg Dosing interval depends on GA: <30 + 0 weeks: 48 hourly 30 + 0–34 + 6 weeks: 36 hourly ≥ 35 + 0 weeks: 24 hourly Concurrent cyclo-oxygenase inhibitors (indomethacin or ibuprofen): Extend dosing interval by 12 h Therapeutic hypothermia: 36 hourly Comments: Subsequent dosing interval is based on gentamicin concentration at 22 h after the administration of the second dose (table provided)	Route: slow IV injection or IV infusion Dose: 5 mg/kg Dosing interval depends on age: Up to 7 days: every 36 h, to be given in an extended interval dose regimen. 7–28 days: every 24 h, to be given in an extended interval dose regimen. Note: BNF for Children provides more information on gentamicin concentrations and dose adjustment in the ‘Monitoring Requirements’ section.	Route: IV Dose: GA < 32 weeks and PNA 0–7 days: 5 mg/kg/48 h GA 32–37 weeks and PNA 0–7 days: 5 mg/kg/36 h Preterms, PNA 1–4 weeks: 4 mg/kg/day Term neonates: 4 mg/kg/day Therapeutic hypothermia: GA ≥ 36 weeks and PNA 0–1 day: 5 mg/kg/36 h Comments: Dose should be adjusted according to plasma levels It is recommended to determine plasma levels before the second dose of gentamicin Empiric use does not necessitate plasma levels determination	Route: IV, IM Dose: GA < 30 weeks and PNA ≤ 14 days: 5 mg/kg/dose every 48 h GA < 30 weeks and PNA ≥ 15 days: 5 mg/kg/dose every 36 h GA 30–34 weeks and PNA ≤ 10 days: 5 mg/kg/dose every 36 h GA 30–34 weeks and PNA 11–60 days: 5 mg/kg/dose every 24 h GA ≥ 35 weeks and PNA ≤ 7 days: 4 mg/kg/dose every 24 h GA ≥ 35 weeks and PNA 8–60 days: 5 mg/kg/dose every 24 h Comments: Determination of dosing interval requires consideration of multiple factors including concomitant medications (e.g., ibuprofen, indomethacin), history of birth depression, birth hypoxia/asphyxia and presence of cyanotic congenital heart disease. Dosage should be individualized based upon serum concentration monitoring. Higher doses and different dosing intervals may be required to achieve target concentrations if MIC ≥ 1 mg/L	Route: IV Dose: PMA ≤ 29 weeks and PNA 0–7 days: 5 mg/kg/48 h PMA ≤ 29 weeks and PNA 8–28 days: 4 mg/kg/36 h PMA ≤ 29 weeks and PNA ≥ 29 days: 4.5 mg/kg/24 h PMA 30–34 and PNA 0–7 days: 4.5 mg/kg/36 h PMA 30–34 and PNA ≥ 8 days: 4 mg/kg/24 h PMA ≥ 35 weeks: 4 mg/kg/24 h Comments: Renal function and drug elimination are most strongly correlated with PMA PMA is the primary determinant of dosing interval, with Postnatal Age as the secondary qualifier.	Route: IV, IM Dose: 5 mg/kg when GA < 44 weeks, thereafter 7 mg/kg Dosing interval: <30 weeks: every 48 h 30–36 weeks: every 36 h Term infants: every 36 h in the first week of life, thereafter every 24 h Comments: Extend the dosing interval if the renal function is poor. It may be helpful to measure the level 24 h after the first dose is given. Individualized treatment: a strategy to individualize treatment in very immature infants (≤ 28 weeks gestation) may be followed by measuring the serum gentamicin level 22 h after a single dose of 5 mg/kg. The timing of the next dose is then calculated according to the level (table provided).

LD = loading dose, MD = maintenance dose, GA = gestational age, PNA = postnatal age, PMA = postmenstrual age (equal to GA + PNA).

## Data Availability

All data collected for this project have been provided in the Appendix A. The corresponding authors can be contacted if additional information is requested.

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
