# Peer review of "Neonatal Drug Formularies—A Global Scope"

_children, 2023, doi:10.3390/children10050848_

Round 1

Reviewer 1 Report

An interesting paper describing and comparing neonatal formularies. My comments are minor:

The paper is long and some of the descriptions of the formularies can be put into a supplementary document (pages 7-12).

Table 2 needs dividing into 2  separate tables - it is too long at present

Table 3 the units need to be the same in each column to allow comparison. At present it is difficult to see the differences. May be better as a diagram

Author Response

An interesting paper describing and comparing neonatal formularies. My comments are minor:

The paper is long and some of the descriptions of the formularies can be put into a supplementary document (pages 7-12).

WE THANK THE REVIEWER FOR THIS SUGGESTION, AND WE HAVE MOVED A SIGNIFICANT PART OF THE FORMER PAGES 7-12 TO ANOTHER SUPPLEMENT

Table 2 needs dividing into 2  separate tables - it is too long at present

WE THANK THE REVIEWER FOR THIS SUGGESTION, AND WE HAVE SUBDIVIDED THE FORMER TABLE 2 INTO DIFFERENT TABLES 2A-2E. ALONG THE SAME LINE, TABLE 3 HAS ALSO BEEN SUBDIVIDED TO TABLE 3A TO 3C.

Table 3 the units need to be the same in each column to allow comparison. At present it is difficult to see the differences. May be better as a diagram

WE THANK THE REVIEWER TO RAISE THIS ISSUE. HOWEVER, WE PREFER NOT TO ADAPT THIS, AS THIS IS ANOTHER BOTTLENECK WHEN COMPARING FORMULARIES AND HAVE ADDED THIS TO THE PAPER AS A RELEVANT OBSERVATION;

s a result of this comparison, have any weaknesses (points that need to be included as information but are lacking) in each formulary been investigated?

WE HAVE FOCUSED ON THE AVAILABILITY AND HAVE COMPARED THE DIFFERENT FORMULARIES, USING A STRUCTURED APPROACH AND ASSESSMENT ON CHARACTERISTICS

Reviewer 2 Report

The manuscript entitled “Neonatal drug formularies – A Global Scope” is recommended for publication in the Children after minor revisions.

Please consider the following suggestions and corrections:

Lines 75, 83, 88, 90, and throughout the text: Add a space between the last word and the reference, e.g., updates [1].

Line 79: Insert a square bracket: Characteristics [2].

Line 96: Do not insert a space: case reports/case series

Line 142: Insert a space before [Document S1].

Line 151: Insert a space before [Document S2].

Line 368: Write the term with a word and put the abbreviation in brackets, for example, intravenous (IV).

Line 559: Table 2 is not clear. Because the columns are too narrow, longer words are split incorrectly. For example, appropriat e, recommendatio n, pharmacogenetic s, Storage/Stabilit y’…  I recommend using a landscape format (page layout) to increase the width of the columns.

When mathematical symbols are used as adjectives, i.e. with one number that is not part of a mathematical operation, do not leave a space between the symbol and the number. E.g.: >3 months

Is the term "Administrat" written upright in the first column in the correct place? It looks like the last letter is missing. Should it be one line up, and "Monitoring" in that place?

Line 617: Table 3 is not transparent. I recommend using a landscape format (page layout) to increase the width of the columns.

Use an en dash (–) to express a range. E.g., 5–10 mg/kg/day

Instead of mcg, use the official SI unit µg or micrograms.

Do not leave a space between the symbol and the number, e.g., (≤28 weeks

Line 619: Add the meaning of the abbreviations IV, IM, and MIC.

Line 840: Correct to D.S., A.S., and K.A.;

Line 842: Correct to D.S., A.S., and K.A.;

Author Response

We thank the reviewer for the overall very supportive assessment, and have considered the editing suggestions. However, the editing of the tables is not fully under own control, so we request the editing office of the journal to further reflect on this and adapt this to improve the readability. 

Reviewer 3 Report

This study collected and compared information on pharmaceutical formularies for newborns by questionnaires.

We have commented on the results below. Please refer to them as a reference.

As a result of this comparison, have any weaknesses (points that need to be included as information but are lacking) in each formulary been investigated?

Introduction

I think it is easy to understand.

Methods

I think it is easy to understand.

Have you investigated how unapproved drugs and off-label drugs are described in each formulary?

Results

I think the way the results were presented was concise and easy to understand.

If you have surveyed textbooks and guidelines (e.g. Nelson's) in the questionnaire, it would be good to add the information in the Appendix.

Author Response

This study collected and compared information on pharmaceutical formularies for newborns by questionnaires. We have commented on the results below. Please refer to them as a reference.

Introduction

I think it is easy to understand.

 Methods

I think it is easy to understand.

 Have you investigated how unapproved drugs and off-label drugs are described in each formulary?

AS THIS IS COMMONLY NATIONAL OR SUPRANATIONAL, WE A PRIORI NEVER HAD THE INTENTION TO QUANTIFY UNAPPROVED OR OFF LABEL DRUGS IN THESE DIFFERENT LEGAL AND REGULATORY REGIONS. ALTHOUGH RELEVANT, WE ARE NOT AWARE OF ANY OF THESE ANALYSES BEYOND NATIONAL OR SUPRANATIONAL TERRITORIES (LIKE EG THE EUROPEAN UNION)

Results

I think the way the results were presented was concise and easy to understand.

If you have surveyed textbooks and guidelines (e.g. Nelson's) in the questionnaire, it would be good to add the information in the Appendix.

WE FELT THAT OUR SEARCH WAS ALREADY QUITE EXTENSIVELY DESCRIBED, AND WE HAVE SOMEWHAT FURTHER ADAPTED THIS DESCRIPTION. IN ESSENCE, THESE SOURCES WERE NOT INCLUDED A PRIORI.